# Innovation as a Factor Increasing Fruit Consumption: The Case of Poland

**DOI:** 10.3390/nu14061246

**Published:** 2022-03-16

**Authors:** Dagmara Stangierska, Iwona Kowalczuk, Katarzyna Widera, Dawid Olewnicki, Piotr Latocha

**Affiliations:** 1Department of Pomology and Horticulture Economics, Institute of Horticulture Sciences, Warsaw University of Life Sciences—SGGW, Nowoursynowska 166, 02-787 Warszawa, Poland; dagmara_stangierska@sggw.edu (D.S.); dawid_olewnicki@sggw.edu.pl (D.O.); 2Department of Food Market and Consumer Research, Institute of Human Nutrition Sciences, Warsaw University of Life Sciences—SGGW, Nowoursynowska 159C, 02-776 Warsaw, Poland; 3Department of Economics, Finance, Regional and International Research, Faculty of Economics and Management, Opole University of Technology, Prószkowska 76, 45-758 Opole, Poland; k.widera@po.edu.pl; 4Department of Environmental Protection and Dendrology, Institute of Horticulture Sciences, Warsaw University of Life Sciences—SGGW, Nowoursynowska 166, 02-787 Warsaw, Poland; piotr_latocha@sggw.edu.pl

**Keywords:** innovation, consumer behavior, fruit market

## Abstract

Due to the low level of fruit consumption in relation to dietary recommendations in many European countries, including Poland, multidirectional actions should be taken to increase the consumption of these products. One of the ideas could be the introduction of innovative products. The main goal of the study is to determine the relationship between consumer propensity to purchase innovative products and the frequency of consumption of fruits and their preserves of consumers. The research sample consisted of 600 respondents who declared to consume fruit and were responsible for food shopping in their households. The results obtained indicate that consumers with a higher propensity to purchase innovative products consumed fruit and fruit preserves more. In addition, statistically significant differences were found between innovators and non-innovators in terms of income, expenditures on fruit purchases, places where fruit and fruit preserves were purchased and product characteristics that determined the purchase decision. The logistic regression results indicate that a higher frequency of supermarket/hypermarket and online shopping, a higher weekly spending on fruit and a greater importance attributed to the biodegradability of the packaging increased the favorability of innovation relatively to fruit products (by 23.8%, 31.4%, 32.7% and 21.6%, respectively). The relationships found may have important implications for both private and public stakeholders in the fruit and vegetable sector.

## 1. Introduction

There is a close correlation between the innovative activity of enterprises and consumers; innovative enterprises (firms that implement innovations), by introducing new or improved products into the market, pique consumers’ interest with their offer, whereas innovative consumers (consumers willing to purchase innovative products), by exerting pressure on companies, motivate them to create innovative solutions [1,2]. The results of studies conducted in recent years on consumer innovation in the food market suggest that product innovation (new or an improved version of previous goods) in a particular market sector can stimulate consumer buying behavior for all products in that sector [3]. This raises the question of whether consumers can be encouraged to increase their consumption of health-beneficial products through innovation, thereby motivating them to behave in line with dietary recommendations. This paper attempts to clarify this issue by analyzing the example of fruit and fruit preserves.

Producers of fruit and fruit preserves offer many diverse, innovative solutions. They fit into the innovation categories selected by XTC World Innovation, such as pleasure, health, physical, convenience and ethics, which reflect consumer expectations regarding the directions for developing the market offer [3].

New varieties of fruit are introduced to the market, with unusual form, flavor and higher nutritional value than traditional fruit [4,5]. The market offer is diversified by visually appealing fruit mixes [5], as well as tasty and convenient fruit snacks [6]. Consumers looking for healthy solutions are offered fruit and preserves sourced with ecological cultivation [7], as well as fruit preserves enriched with additional nutrients [8]. Innovation also applies to packaging, such as the so-called active packaging, which keeps the fruit fresh, nutritious and appealing for longer, while making it more convenient to store and consume the product [9]. An important motivation for producers to constantly search for innovations in the agriculture–food market is the high market failure rate of new products, which is mainly due to the consumers’ lack of acceptance of such products [10].

According to researchers studying the issue of innovation in the consumer goods market, both today and in recent decades, the key to the effective commercialization of an innovative offer is recognizing the consumers’ needs and their reaction to new products, as well as characterizing the recipients of the new products [11]. Over the years, many concepts of the innovation process have been proposed [12,13]. One of the most prominent theories explaining this matter is Rogers’ model [14]. Rogers identifies five recipient groups based on their openness to innovation, namely, innovators, early adopters, early majority, late majority and laggards, with the corresponding shares in the overall population of 1.5%, 13.5%, 34%, 34% and 16%. In order to learn about consumer behaviors towards market innovations, researchers typically conduct a purchase-intention study and analyze opinions on new products [15], while, when characterizing consumer innovation, socio-demographic and psychographic features are analyzed [16]. As for the socio-demographic features, it has been proven that the innovation level is affected by age (younger consumers are more innovative) [17,18], level of education [19,20], income [19,21] and country of origin [22]. The innovation level is also conditioned by features such as openness to new experiences, curiosity and susceptibility to external influence [16,23], including media [24,25] and influencers [26]. Research on consumer innovation in the fruit market has shown that young and middle-aged consumers with higher education and income are more open to new products. It has also been shown that innovations following the trends of health and ethics (environment-friendly innovations) receive the most attention from recipients [27].

Actions to promote a healthy lifestyle and prevent diseases in most European countries focus on initiatives oriented at increasing the consumption of fruits and vegetables among various consumer groups [28]. A diet rich in fruit and vegetables is widely recommended due to the health benefits of these products [29]. Numerous research studies have proven that consuming fruit and fruit preserves is beneficial to the prevention of some chronic diseases [30], including type 2 diabetes [31], obesity [32], cardiovascular diseases [33], hypertension [34], various types of cancer [35,36] asthma [37], depression [38] and cognitive disorders [39]. In addition to having a positive effect on the health of individuals and the general population, switching to a plant-based diet can also have a significant impact on the environment by reducing the carbon footprint [40]. According to experts, eating at least 400 g of fruits and vegetables per day can have the most beneficial effects for personal well-being and the planet [41,42]. However, despite many initiatives to promote healthy lifestyles, the populations of less than half of the WHO member states consume fruits and vegetables according to the WHO recommendations [43,44]; Poland also belongs to this group.

According to Statistics Poland, in 2020, the consumption was merely 46.3 kg/year [45], which amounts to 127 g per day (293 g, including vegetables). This is only 73% of the daily consumption recommended by the WHO [46]. According to the Eurostat data from 2019, only 62.5% of Poles eat fruit at least once per day (EU average—67%) and women consume fruit more often than men do (72.9% and 56.4%, respectively) [47].

Considering the too-low fruit consumption (in relation to dietary recommendations) both in Poland and other countries and the potential relationship, not yet confirmed by existing studies, between consumer tendencies for innovative behaviors relative to the fruit market and their consumption of these products, this research study was undertaken to determine the following:-Consumer structures based on affinity for innovation in the fruit and fruit preserve market;-Characteristics of consumers with varying tendencies for innovative behaviors;-Correlation between the consumer tendency to buy innovative fruit and fruit preserve products and the level of consumption of these products and expenses incurred to purchase them;-Features of an innovative offer and means to distribute it that would stimulate consumer interest in the innovative offer.

## 2. Materials and Methods

### 2.1. Study Design and Participants

The paper is based on the results of a questionnaire research study conducted with the CAWI method by a professional research company, BioStat; this methodological choice guaranteed the ethical standards necessary for the execution of the study. The ethical aspects followed throughout the study ensured the continued safety of participants, as well as the integrity of the accumulated data. A brief description of the study and its aim, and the declaration of anonymity and confidentiality were given to the participants before the start of the questionnaire. Respondents did not provide their names nor contact information (including the IP address) and could finish the survey at any stage. The answers were saved only when participants clicked the “submit” button after filling in the questionnaire.

The online survey was conducted in full observance of the national and international regulations compliant with the Declaration of Helsinki (2000). The personal information and data of the participants were anonymous, according to the General Data Protection Regulation of the European Parliament (GDPR 679/2016). The survey did not require approval by the ethics committee because of the anonymous nature of the online survey and impossibility of tracking sensitive personal data.

Study participants were recruited among the people registered with the respondent panel of the BioStat research company. Respondents were non-randomly selected for the study—they were adults who declared to eat fruit at least once per month and were responsible or co-responsible for buying fruits and fruit preserves in their household. Ultimately, the criteria assumed for selection were met by 600 people.

### 2.2. Questionnaire

The research study was conducted with the use of an original questionnaire. In order to specify questions and clarify any ambiguities, prior to the study proper, a pilot study was carried out; the questionnaire was distributed to a group of 30 people, together with a form allowing respondents to assess the questionnaire layout, comprehension of the questions asked and relevance of the questions for the goal of the study. The questionnaire was constructed using E. Rogers’ scale [14] and the scales developed for nutrition research validated and approved by the Scientific Research Committee of the Polish Academy of Sciences (KomPAN^®^)Warsaw, Poland [48].

According to the assumed goal of the study and the formulated research problems, the survey questionnaire included questions regarding the following:-Reactions of the respondents to innovative products in the fruit market (answers on a scale of 1–5, where 1—“I buy new products immediately after they show up on the market”; 2—“I buy new products relatively quickly, though after some consideration”; 3—”I buy new products when some of my acquaintances have tried them and given positive opinions”; 4—“I buy new products when most of my acquaintances have tried them and given positive opinions”; and 5—“I am reluctant to buy new products”);-Frequency of eating fruits and preserves (answers on a scale of 1–7: 1—never; 2—less often than once a month; 3—1–3 times per month; 4—once a week; 5—several times a week; 6—once a day; and 7—several times a day);-Places for buying fruits and fruit preserves (answers on a scale of 1–6: 1—never; 2—less than once per month; 3—1–3 times per month; 4—once a week; 5—several times a week; and 6—once a day);-Weekly expenses on fruits and fruit preserves (single choice question with 5 ranges of expenses incurred);-Factors conditioning purchase decisions in the fruit market (position scale 1–7: 1—definitely irrelevant factor; 7—definitely relevant factor);-Respondents’ characteristics (accounting for gender, age, place of residence, education and income).

### 2.3. Characteristic of Respondents

In terms of gender, the sample consisted of 52% women and 48% men. The age structure of the studied group was: 18–19 years old—18.3%; 30–44 years old—29.5%; 45–59 years old—23.3%; 55 and older—28.3%. Nearly 40% of the respondents lived in rural areas, 32% in towns of under 100,000 people, 16.8% in towns with 100–500 thousand residents and 11.3% in cities with a population larger than 500,000 people. In terms of education, the largest group was people with secondary education (34.7%); a total of 29.2% of the respondents had completed vocational education, 28.3% had completed higher education and only 7.8% had completed primary education. Among the respondents, most lived in households of 3–4 people (55.3%). The analysis of the economic situation of the respondents showed that nearly half (48.8%) of them had a monthly income of PLN 1500–3000 per person, while 20% made less than PLN 1500; a total of 14.8% of the respondents had an income in the range of PLN 3001–4500 and 10.8% earned more than PLN 4500 per person per month (Table 1).

### 2.4. Statistical Analyses

Based on the questions in the survey, the variables to be subjected to a later analysis were identified. The rank scale (qualitative) of most of the variables accepted for the study determined the choice of adequate statistical tests. The only quantitative variable did not show a normal distribution. Thus, non-parametric Pearson *χ*^2^ and Mann–Whitney U tests were used in the analysis. The statistical significance of differences between selected groups of respondents was tested with the Mann–Whitney U test, whereas the statistical dependence between variables was assessed with the *χ*^2^ test. Descriptive statistics included calculations of relative frequencies (the number of categories of variables expressed as a percentage) and mean, median, minimum and maximum values of the examined variables. The assumed minimum level of significance for all statistical tests was 0.05.

All factors differentiating at the established level of significance between the behaviors of innovators and non-innovators were included as potential independent variables in the logistic regression model. Finally, the model presented below only includes those independent variables whose structural parameters met the condition of statistical significance.

The interpretation of the results (odds ratio) of the logistic regression analysis consisted of determining by what percentage the likelihood of changing consumer behavior shifted with the changes in the value of a specific independent variable.

All calculations were made using the Statistica 14.1 statistical packageunder statsoft license available for university employees.

## 3. Results

### 3.1. Respondents’ Innovation in the Fruit and Preserve Market

In the studied population, 13.0% of the respondents declared to buy new fruits and preserves immediately after they showed up on the market (innovators); a total of 44.8% responded that they purchased new products relatively quickly, though after some consideration (early adopters); a total of 23.0% of the respondents said they bought novelties after some of their acquaintances had tried them and given positive opinions (early majority); a total of 9.8% of the questioned people purchased new products after most of their acquaintances had tried them and given positive opinions (late majority); and 9.0% declared reluctance to buy new products (laggards). Upon projecting the acquired data onto Rogers’ model of distribution [14], the studied population was found to consist mostly of innovators and early adopters, with a significantly lower percentage of those consumer groups who were less enthusiastic or downright skeptical towards innovative products (Figure 1).

Considering the acquired results, for the purpose of further analyses, the studied population was divided into two groups based on innovation level, the group of innovators, including innovators and early adopters (*n* = 349, 58.2%); and the group of non-innovators, consisting of early majority, late majority and laggards (*n* = 251, 41.8%).

### 3.2. Comparative Characteristics of Innovators and Non-Innovators Accounting for Demographic, Social and Economic Features

The conducted analyses showed no statistically significant correlations between consumer groups with different affinities for innovative behaviors and their gender, age, place of residence, number of people in the household or education level.

In the case of income, a larger percentage of innovators than non-innovators declared earnings in the ranges of PLN 1501–3000 and PLN 3000–4500, whereas nearly twice as many non-innovators compared with innovators declared earning less than PLN 1500. The value of the *χ*^2^ = 14.2845 test statistics and the value of *p* = 0.0064 assigned to it indicated a statistically significant correlation in terms of the income levels of innovators and non-innovators. Higher income meant a higher percentage of people with affinity for innovation (Table 2).

### 3.3. Frequency of Consuming Fruit and Fruit Preserves of Innovators and Non-Innovators

Both the innovators and those respondents who were less willing to purchase innovative products most often consumed fruit juices (means of 4.87 and 4.48, respectively) and fresh fruit (4.66 and 4.34). Much less popular among both groups were products such as dried fruit (3.21 and 2.90), fruit and vegetable juices (3.12 and 2.80) and canned fruit (3.04 and 2.59), while the least frequently consumed products, among both innovators and non-innovators, were frozen fruit (2.67 and 2.44), fruit mousses (2.96 and 2.39), fruit/fruit and vegetable salads (2.90 and 2.37), fruit chips (2.54 and 2.02) and freeze-dried fruit (2.12 and 1.69). For all the product categories analyzed, the innovators exhibited a higher level of consumption (Figure 2; detailed data are provided in Appendix A, Table A1).

Regarding fresh fruit and traditional fruit preserves (juices, dried fruit, frozen fruit), differences in the frequency of consumption of these products were statistically significant at *p* < 0.05, whereas in regard to modern fruit preserves (fruit mousses, salads, fruit chips and freeze-dried fruit), they were significant at *p* < 0.001 (Table 3).

### 3.4. Expenses for Purchasing Fruit Incurred by Innovators and Non-Innovators

The majority of innovators (57.6%) declared spending more than PLN 41 (EUR 8.9) per week on fruit, whereas 62.9% of non-innovators spent less than that (Figure 3). The value of the Mann–Whitney U test (5.1779; *p* = 0.0000) indicated a statistically significant difference in expenses on fruit incurred by innovators and non-innovators.

### 3.5. Places for Innovators and Non-Innovators to Purchase Fruits and Fruit Preserves

Both innovators and non-innovators most frequently bought their fruit at discount stores (means of 3.98 and 3.96, respectively) and supermarkets/hypermarkets (3.45 and 3.06). Less popular places for purchasing included convenience stores (3.21 and 3.01), marketplaces (2.96 and 2.76) and local grocery stores, while the least frequently used sources of fruits were street stalls (2.01 and 1.7) and online shopping (1.58 and 1.29) (Figure 4; detailed data are provided in Appendix A, Table A2).

Statistically significant differences between innovators and non-innovators in the frequency of purchasing from the analyzed sources were found in the case of supermarkets/hypermarkets and online shops (*p* < 0.001), while for the local grocery stores, street stalls and marketplaces, the level of statistical significance of the differences was less than 0.05. No differences were found among the consumer groups in the frequency of buying at discount stores and convenience stores (Table 4).

### 3.6. Relevance of Selected Features of Fruits and Fruit Preserves for Innovators and Non-Innovators

Consumers with both high and low innovation levels considered the following features to be the most relevant in their choice of fruit and fruit preserves: freshness (means of 6.40 and 6.61, respectively), taste preferences (5.95 and 6.12) and appearance (5.93 and 6.17). Less important factors included the following: habits (5.20 and 5.45), price (4.90 and 5.24), information on the packaging (4.83 and 4.51), packaging size (4.61 and 4.56), country of origin (4.35 and 4.03) and biodegradability of the packaging (4.09 and 3.50) (Figure 5; detailed data are provided in Appendix A, Table A3).

The innovators, with statistical significance, found features of fruits and fruit preserves such as biodegradability of the packaging (*p* < 0.001), information on the packaging (*p* < 0.05) and country of origin of the products (*p* < 0.05) to be the most relevant, whereas for non-innovators, the importance of price (*p* < 0.05) and habits of consuming specific types of fruit (*p* < 0.05) was higher (Table 5). 

### 3.7. Using Logistic Regression to Analyze the Factors Determining Consumer Innovation

Among the factors stimulating consumer affinity for innovation in the fruit and fruit preserve market accounted for in the regression analysis, statistically significant differences (*p* < 0.05) were found in the case of six of the analyzed independent variables, such as frequency of making purchases in supermarkets/hypermarkets (1) and via the Internet (2); level of weekly expenses on fruit (3); importance of the price (4); biodegradability of the packaging (5); and habits (6) of buying fruits and fruit preserves.

The results of the regression show that a higher frequency of buying at supermarkets/hypermarkets and online increased the chance for a consumer to have an affinity for innovative solutions in the fruit and fruit preserve market by 23.8% and 31.4%, respectively. Higher weekly expenses on fruit resulted in innovation being increased by 32.7% and greater importance of biodegradability of the packaging increased affinity for innovation by 21.6%. On the other hand, a greater importance of price in purchasing fruit and preserves resulted in the chance for innovation to be decreased by 11.8%, whereas a greater significance of habits reduced affinity for innovation by 19.7% (Table 6).

## 4. Discussion

The problem of low fruit and vegetable consumption in comparison with dietary recommendations concerns more than a half of the WHO countries, mainly Eastern Europe [43]. Considering the nutritional value of these products, their protective effects against various chronic diseases and the fact that a diet rich in vegetables and fruit has a beneficial effect on the environment, many countries are undertaking intervention activities, mainly of educational [43] and marketing nature, but also within the scope of the so-called nudge interventions [50]. Such activities are also undertaken in Poland. However, they do not solve the problem, so other methods for stimulating consumer interest in the consumption of these products should be searched for. In the undertaken research study, it was decided to check whether innovation, in its broadest sense, could constitute such a method.

One of the assumed objectives of the study was to determine the consumer structure based on affinity for innovation in the fruit and preserve market. The obtained results indicate that the distribution of the studied population differed from the Rogers’ model distribution in this matter; a much larger percentage of innovators (13.33% and 2.50%, respectively) and early adopters (44.84% and 13.50%) were found, as well as significantly fewer consumers exhibiting the behaviors of early majority (23% and 34%, respectively), late majority (9.83% and 34%) and laggards (9% and 16%). Previous research on this issue also proves a greater consumer affinity for innovative behaviors in the food market [51]; moreover, Gonera et al. [52] have found a higher degree of innovation among consumers with high acceptance of plant-based products. Winger and Wall [53] explain the greater consumer affinity for innovation in the food market with lower risk being related to purchasing innovative products. The risk level related to purchasing decisions regarding new products depends, among other factors, on the extent to which they differ from what is familiar to the consumer [54]; since the majority of food innovations are incremental changes (continuous innovations), the innovative offer does not differ dramatically from traditional products, which lowers the risk and increases the consumer willingness to purchase.

By analyzing the socio-demographic profile of innovators and non-innovators, as opposed to other research studies [17,18,19,20], no differences between the two groups were found in regard to age, sex, education, place of residence or number of people in the household. However, a statistically significant influence of income on the respondents’ innovation level was discovered, confirmed by other research studies, both on Polish consumers [51] and other nationalities [19].

The crucial issue in the conducted study was to determine the differences in the frequency of consuming fruit and preserves between innovators and non-innovators. The obtained results show that all the analyzed product categories were consumed by innovators more frequently, with statistical significance. The most obvious explanation of the fact that innovators consumed fruit and preserves more often is the higher income declared by the members of this group, as well as the correlation between income level and volume of fruit and preserve consumption, which has been proven in earlier research [55,56]. However, in explaining this correlation, it can be assumed with high probability that affinity for innovative behaviors goes hand in hand with seeking information on new products, consequently obtaining knowledge about properties of fruits and preserves, their nutritional value and health benefits (this thesis has been proven in a study on ecological food) [57]. As a result, an innovative consumer becomes convinced that consuming fruit and preserves is useful, which, according to the theory of planned behavior [58], is one of the factors determining their buying intentions and decisions to purchase. The more frequent consumption of fruit and preserves, as well as the higher income declared by the innovators, resulted in them spending more on fruit.

An analysis of the variation in innovators’ and non-innovators’ preferred places for purchasing fruit and preserves only showed statistically significant differences in the case of supermarkets/hypermarkets and online shopping; innovators declared using both of these forms of distribution more often. In relation to both of these places of purchase, the identified difference can be explained by a relatively larger offer of innovative fruit and preserves than in other stores; in the case of supermarkets/hypermarkets this would be the result of a broad selection of products on offer [59], whereas in the case of online stores, of a highly specialized offer [60,61]. Moreover, in the big-box stores, the presence of other customers enhances the bandwagon effect and reduces social risk [62], which might cause the innovative offer to garner more attention. Online stores, on the other hand, allow one to obtain information about the purchased products, which is important for innovative consumers [63,64] and, at the same time, caters to their openness to new experiences and their aspiration to take advantage of innovative solutions in different areas of activity [16].

Both the consumers with high and low levels of innovation found the following factors the most important in selecting fruits and preserves: freshness, preference for taste and appearance. The importance of these factors in selecting food products has also been found in other studies [65,66,67,68,69,70,71]. Previous studies on the consumer-preferred characteristics of fruits and processed fruits have also found the importance of other characteristics of these products, such as health benefits, attractiveness and uniqueness (for tropical fruits), [72], health benefits and convenience (for dried fruits) [73], composition and origin (for canned fruits) [74] or naturalness (for fruit juices) [75]. Differences between innovators and non-innovators in evaluating the importance of determinants for selecting fruits and preserves were found in the case of factors such as biodegradability of the packaging, information included on the packaging and country of origin, which were more important to the innovators; price and habits were more important to non-innovators. More innovative consumers have also been found to value environment-friendly and healthy solutions in the study by Samoggi and Nicolodi [27]. This correlation can be justified by a (psychologically conditioned) greater openness of this consumer group to innovative solutions [16], as well as greater awareness of the benefits coming with those. Moreover, current research suggests that, in the case of fresh fruit, biodegradability of the packaging enhances the innovative image of a product and positively affects its selection [76,77]. As for innovators paying attention to the information on the packaging and the product’s country of origin, these can be linked to a greater affinity for seeking knowledge about purchased goods, which is characteristic of this consumer group [78]. When it comes to those features of fruits and preserves that are more important to non-innovators, the greater significance of price should be attributed to those with lower incomes in this consumer group and the resulting limitations regarding the selection of purchased products [79], whereas sticking to buying habits is considered to be a typical feature of consumers reluctant to adopt innovative behaviors [16,80].

## 5. Strengths, Limitations and Future Research

The obtained results can be important both for the enterprises of the fruit–vegetable sector and for the institutions and organizations dealing with nutrition. Studying consumer behaviors and expectations in regard to innovative products can be helpful to create marketing strategies for such products and positively affect their adaptation and diffusion, eventually contributing to a greater consumption of fruit and fruit preserves.

The weakness of this study could be the relatively small sample group, though the criteria for its selection (consuming fruit and preserves at least once per month and being responsible for buying these products for the household) could be seen as an explanation of the final number of participants of the study. The research study was indubitably limited by the fact that the sample consisted exclusively of Polish consumers, which calls for confirming the observed correlations with studies in other countries. It would also be advisable to make future research more detailed by analyzing more factors that potentially differentiate consumer behaviors, or by focusing the analysis on specific types of products. It should also be noted that market behaviors of consumers eating fruit less often than once per month also need to be studied; this was omitted in this research study.

## 6. Conclusions

The results of the study show that consumers with greater affinity for purchasing innovative products ate fruit and fruit preserves more often. Differences were found between innovators and non-innovators in terms of income, expenses incurred for buying fruits and places for purchasing fruit and fruit preserves, as well as product features determining the decision to buy.

The regression analysis showed that selling innovative products through modern channels of distribution, using biodegradable packaging and rationalizing the prices of the innovative offer showed to be the most promising factors in terms of affecting the increase in consumer affinity for innovative behaviors.

## Figures and Tables

**Figure 1 nutrients-14-01246-f001:**
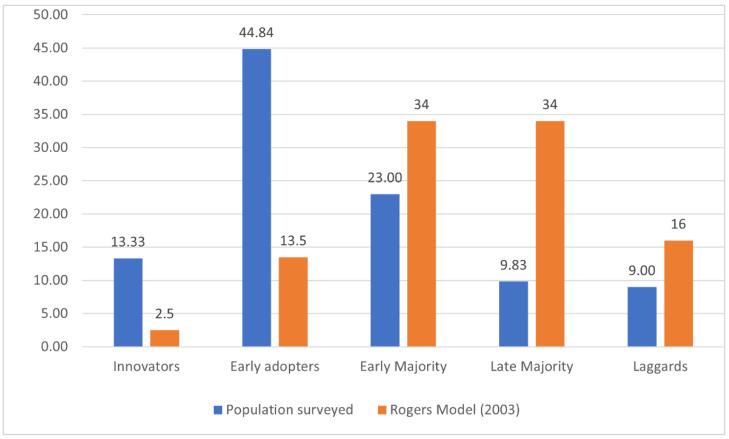
Comparison of the studied population’s distribution with Rogers’ model, accounting for innovation level (%).

**Figure 2 nutrients-14-01246-f002:**
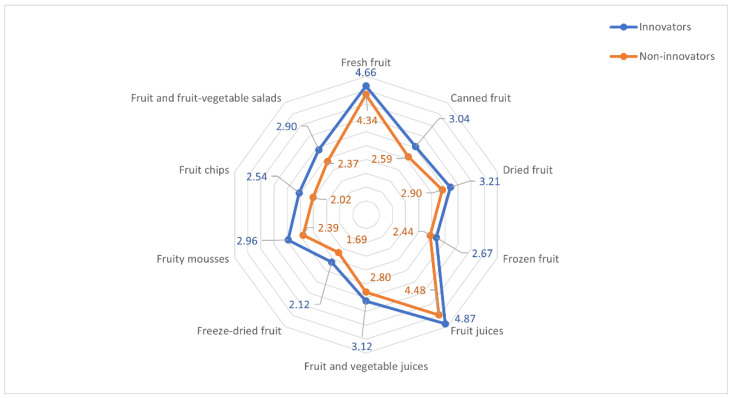
Frequency (On a scale of 1–7: 1—never; 2—less often than once a month; 3—1–3 times a month; 4—once a week; 5—several times a week; 6—once a day; 7—several times a day)of consuming fruits and fruit preserves of innovators and non-innovators.

**Figure 3 nutrients-14-01246-f003:**
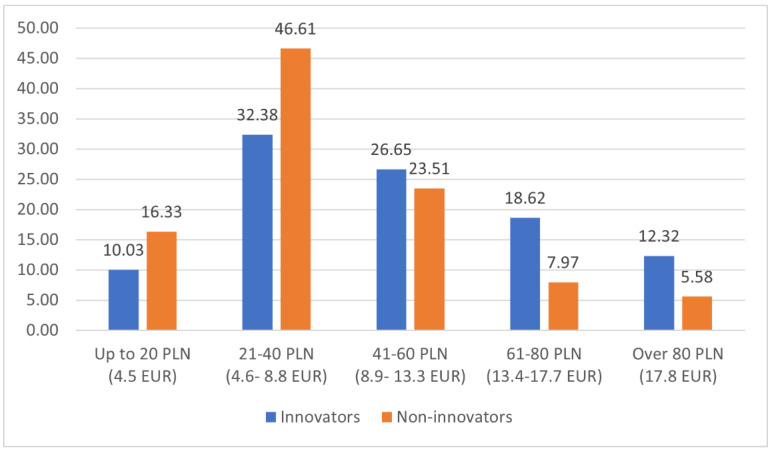
Weekly expenses on fruits and fruit preserves by innovators and non-innovators.

**Figure 4 nutrients-14-01246-f004:**
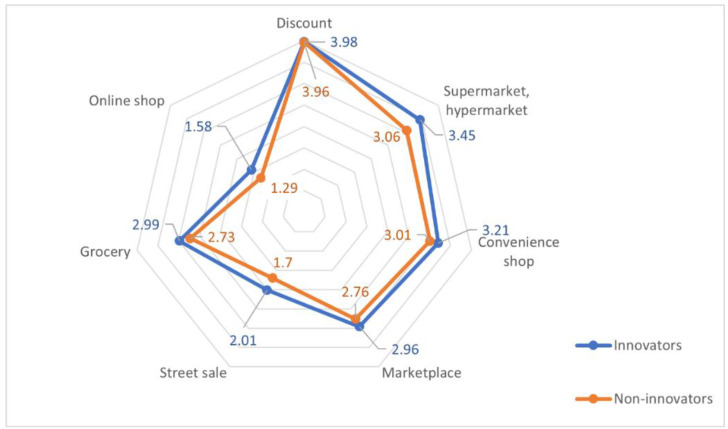
Frequency (On a scale of 1–6: 1—never; 2—less than once a month; 3—1–3 times per month; 4—once a week; 5—several times a week; 6—once a day) of purchasing fruit and fruit preserves at selected places of purchase of innovators and non-innovators.

**Figure 5 nutrients-14-01246-f005:**
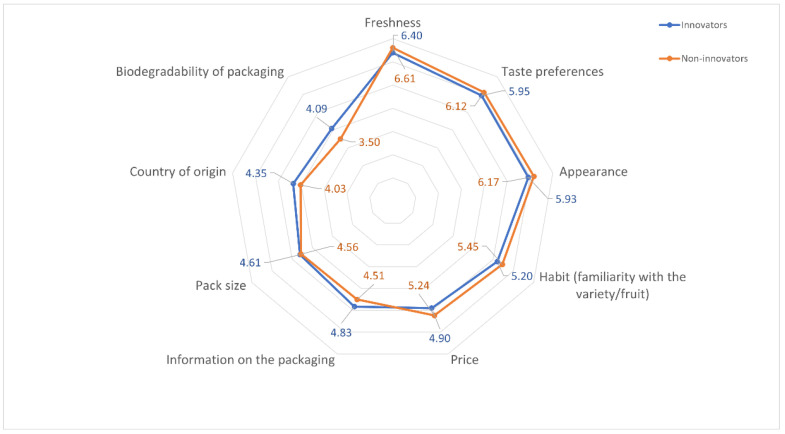
Significance (On a scale of 1–7: 1—definitely irrelevant factor; 7—definitely relevant factor) of selected features of fruits and fruit preserves for innovators and non-innovators.

**Table 1 nutrients-14-01246-t001:** Sample characteristics (%).

**Gender**									
Female					Male				
52.00					48.00				
**Age**									
18–29		30–44			45–59			Over 55	
18.33		29.50			23.33			28.33	
**Place of Residence**									
Rural areas		Towns, up to 100,000 residents			Towns, 100,000–500,000 residents			Cities, over 500,000 residents	
39.83		32.00			16.84			11.33	
**Education**									
Primary		Vocational			Secondary			Higher	
7.83		29.17			34.67			28.33	
**Number of People in the Household**									
1–2			3–4				5 and more		
29.83			55.33				14.84		
**Per Capita Income PLN (EUR) ***									
Under 1500 (332.6)	1500–3000 (332.7–665.2)			3001–4500 (665.3–997.8)		4501–6000 (997.9–1330.4)			Over 6000 (1330.5)
20.17	48.83			14.83		6.17			4.67

* As of 19 January 2022 [49].

**Table 2 nutrients-14-01246-t002:** Characteristics of innovators and non-innovators accounting for demographic, social and economic features.

Variable	Innovators (349)	Non-Innovators (251)	Statistic	*p*-Value
Age (years)				
Average (median)	46 (45.8)	47 (46.10)	Z = −0.2671 *	0.7893
Gender (%)				
Female	50.14	54.58	*χ*^2^ = 1.1523	0.2831
Male	46.86	45.42		
Place of Residence (%)				
Rural areas	40.97	38.24	*χ*^2^ = 0.8133	0.8463
Towns, up to 100,000 residents	30.95	33.47		
Towns, 100,000–500,000 residents	16.33	17.53		
Cities, over 500,000 residents	11.75	10.76		
Education (%)				
Primary	6.30	9.96	*χ*^2^ = 7.1373	0.0677
Vocational	32.66	24.30		
Secondary	34.67	34.66		
Higher	26.36	31.08		
Number of People in the Household (%)				
1–2	29.23	30.68	*χ*^2^ = 1.4869	0.4755
3–4	54.44	56.57		
5 and more	16.33	12.75		
Monthly Per Capita Income (%)				
Under PLN 1500 (332.6 EUR)	16.92	28.89	*χ*^2^ = 14.2845	0.0064
1501–3000 (332.7–665.2 EUR)	51.96	48.44		
3001–4500 (665.3–997.8 EUR)	18.13	12.89		
4500–6000 (997.9–1330.4 EUR)	8.16	4.44		
Over PLN 6000 (1330.5 EUR)	4.83	5.33		

* Z-statistics and the corresponding *p*-values refer to the comparison of the medians with a non-parametric Mann–Whitney U test.

**Table 3 nutrients-14-01246-t003:** Variation in frequency of consuming fruits and fruit preserves for innovators and non-innovators.

	Variable	Z-Statistic *	*p*-Value *
	Fresh fruit	3.1742	0.0015
Traditional fruit preserves	Dried fruit	2.9140	0.0036
	Frozen fruit	2.6022	0.0093
	Fruit juices	3.2152	0.0013
	Fruit and vegetable juices	2.6848	0.0073
Modern fruit preserves	Freeze-dried fruit	4.6026	0.0000
	Canned fruit	4.3668	0.0000
	Fruit mousses	5.4274	0.0000
	Fruit chips	5.0225	0.0000
	Fruit and fruit-vegetable salads	4.7406	0.0000

* Z-statistics and the corresponding *p*-values refer to the comparison of the medians with a non-parametric Mann–Whitney U test.

**Table 4 nutrients-14-01246-t004:** Variations in the frequency of buying fruits and fruit preserves from selected sources for innovators and non-innovators.

Variable	Z-Statistic *	*p*-Value *
Discount	0.0843	0.9328
Supermarket, hypermarket	4.1529	0.0000
Convenience store	1.7903	0.0734
Marketplaces	2.0570	0.0397
Street stall	2.8337	0.0046
Grocery store	2.7099	0.0067
Online shop	3.7700	0.0000

* Z-statistics and the corresponding *p*-values refer to the comparison of the medians with a non-parametric Mann–Whitney U test.

**Table 5 nutrients-14-01246-t005:** Variations in significance of selected features of fruit and fruit preserves for innovators and non-innovators.

Variable	Z-Statistic *	*p*-Value *
Price	−2.6263	0.0086
Appearance	−1.8090	0.0705
Freshness	−1.4786	0.1393
Taste preferences	−0.6476	0.5172
Country of origin	2.0305	0.0423
Packaging size	0.0781	0.9378
Information on the packaging	2.3897	0.0169
Biodegradability of the packaging	4.0720	0.0000
Habits (familiarity with the variety/fruit)	−2.1706	0.0300

* Z-statistics and the corresponding *p*-values refer to the comparison of the medians with a non-parametric Mann–Whitney U test.

**Table 6 nutrients-14-01246-t006:** Values of logistic regression model coefficients.

Variable	Coefficient	Odds Ratio	Standard Error	*t*-Stat. (593)	*p*-Value
Frequency of buying at super-/hypermarkets	0.213	1.238	0.081	2.639	0.009
Frequency of buying online	0.273	1.314	0.119	2.294	0.022
Expenses on fruit	0.283	1.327	0.084	3.366	0.001
Importance of price in buying fruit	−0.126	0.882	0.060	−2.094	0.037
Importance of biodegradability of the packaging in buying fruit	0.196	1.216	0.053	3.698	0.000
Importance of habits in buying fruit	−0.220	0.803	0.072	−3.041	0.002
Constant	−0.42				

## Data Availability

The authors confirm that the datasets analyzed during the study are available from the corresponding author upon reasonable request.

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
