# Peer review of "Innovation as a Factor Increasing Fruit Consumption: The Case of Poland"

_nutrients, 2022, doi:10.3390/nu14061246_

Round 1

Reviewer 1 Report

  • In this study, what do you mean “innovation” or “innovative products”? It is not easy to imagine the concrete examples for non-specialized readers.
  • Section 5.1 How should we interpret the fact that the composition ratio of respondents’ innovation level is different with the Rogers model.
  • In this study, the questions of the survey for two respondents groups are mainly about the consumers’ general purchasing and consumption behavior of. Therefore, there seems to be only an indirect discussions or guess about the impact of innovation on such behaviors. Could you have asked respondents more direct questions about how innovation affects consumers’ behavior and their thinking?

Author Response

Thank you very much for all your valuable comments. We have made appropriate corrections in the text of the article. We have addressed all comments in the attached document.   We hope that the changes we have introduced are sufficient. If not, we would appreciate further suggestions.

Reviewer 2 Report

The manuscript entitled „Innovation as a factor increasing fruit consumption. The case of Poland.” presents interesting issue, but some issues should be corrected.

General:

Authors should prepare their manuscript according to the instructions for authors.

It seems that none of the authors is native English speaker, so some sentences are even hard to follow (e.g. “The main focus of the study was to learn about [consumer] preferences and market behavior”). Authors should have their manuscript corrected by the professional English correcting agency.

Authors should refer international references published in English, instead of some national references which are not available for international readers (e.g. Ref. 25 by Greobwic)

Abstract:

The aim of the study should be clearly formulated.

The studied group should be described (its basic characteristics)

Authors should present specific results of the study accompanied by results of the statistical analysis.

Introduction:

Authors should avoid multiple references, as referring 82 (!) of them in introductory sections is excessive.

Instead of presenting separately Introduction, Theoretical Background, and Goals, Authors should briefly present Introduction (1 page and about 20 references would be enough) to properly justify the study

Authors failed to justify the need for their study – they should present what is already known and what are the “gaps” in the scientific knowledge to formulate the aim of their study.

Materials and Methods:

The study is conducted in a group of human subjects, and their personal data are obtained (age, gender). In the presented manuscript, there was no information about ethical committee agreement, so it is not specified if Authors did not obtain such agreement, or obtained, but did not provide information about it. It must be indicated as a serious ethical problem, that Author probably did not obtain ethical committee agreement for his study.

There is a serious problem with the applied scale of frequency, as there are some frequencies which are not represented, as Authors used the following scale: “never, several times a year, 1-3 times per month, once a week, 2-3 times a week, once a day, and 2-3 times a day”. And what about consuming 4 times a week? what about consuming 5 times a week? what about consuming 6 times a week? what about consuming more often that 3 times a day?

The presented data are not consistent – Authors stated that “Respondents were non-randomly selected for the study - they were adults who declared eating fruit at least once per month” – if so, how is it possible that they asked about eating fruit in their questionnaire and some respondents answered that they consume fruits never, or several times a year. It seems that Authors do not present real information about including respondents to the study.

Inclusion and exclusion criteria, as well as qualifying procedure (flow chart) should be presented.

It seems that Authors classified respondents according to Rogers’ model, based on the own declaration of respondents, which is a serious bias of the study.

In the presented study there is almost no clear methodology presented, so we do not know what was done and how – Authors should precisely present the detailed methodology of their data analysis.

It seems that Authors did not verify the normality of distribution of their data – they should do it and present the related methodology.

After verifying the normality of distribution, in case of parametric distribution mean ± SD should be presented, while for nonparametric distribution – median accompanied by minimum and maximum value.

The applied statistical test should be based on distribution

Results:

It seems that Authors did not verify the normality of distribution of their data – they should do it and present the related methodology.

After verifying the normality of distribution, in case of parametric distribution mean ± SD should be presented, while for nonparametric distribution – median accompanied by minimum and maximum value.

The applied statistical test should be based on distribution

Authors should present tables instead of figures, to be easier to follow by readers

Cut-offs for per capita income should be justified based on a proper reference

Discussion:

Authors should present issue of fruit purchase in the aspect of buying them for specific consumers, e.g. mothers buying for children, or older members of the family (is innovativeness the only determinant?)

Authors should present issue of choice of specific fruit products – what products are chosen?

Authors should base their discussion of a proper national and international perspective

Conclusions:

Authors should briefly (2-3 sentences are enough) present major conclusions directly from their study

Authors’ Contributions:

Authors should properly define the contributions – e.g. what do Authors mean by “validation” if they did not conduct any validation in their study?

It seems that contribution of some Authors was only minor and they did not participate in preparing manuscript. There is a serious risk of a guest authorship procedure which is forbidden. In such case (if they did not participate in manuscript preparation in any way) they should be rather presented in Acknowledgements Section and not be indicated as authors of the study.

Author Response

(The authors gave the same response as above.)

Round 2

Reviewer 2 Report

The manuscript entitled „Innovation as a factor increasing fruit consumption. The case of Poland.” presents interesting issue, but some issues should be corrected. Unfortunately my major comments were ignored.

General:

Authors should refer international references published in English, instead of some national references which are not available for international readers (e.g. Ref. 3 by Greobwic)

Abstract:

Authors should present specific results of the study accompanied by results of the statistical analysis.

Introduction:

Authors should avoid multiple references, as referring 58 (!) of them in introductory sections is excessive.

Authors should briefly present Introduction (1 page and about 20 references would be enough) to properly justify the study

Authors failed to justify the need for their study – they should present what is already known and what are the “gaps” in the scientific knowledge to formulate the aim of their study.

Materials and Methods:

The study is conducted in a group of human subjects, and their personal data are obtained (age, gender). In Authors did not obtain ethical committee agreement. It must be indicated as a serious ethical problem. Authors should not refer the other study which did not obtain such agreement, as each situation is individual and Authors should not reproduce approach from the other study.

There is a serious problem with the applied scale of frequency. Authors responded my point, but it is impossible to verify it, as it seems that Authors referred random references not associated with formulated sentences – for the position of “the Scientific Research Committee of the Polish Academy of Sciences (KomPAN) [59,60]”, the following reference is presented: [59] - WHO Global Conference on Air Pollution and Health; Geneva, Switzerland – I have no idea how is this nutritional questionnaire associated with WHOs conference on global pollution…

Inclusion and exclusion criteria, as well as qualifying procedure (flow chart) should be presented.

It seems that Authors classified respondents according to Rogers’ model, based on the own declaration of respondents, which is a serious bias of the study. Authors indicate 2 studies which applied such approach, but none of them is relevant for the discipline.

In the presented study there is almost no clear methodology presented, so we do not know what was done and how – Authors should precisely present the detailed methodology of their data analysis.

Authors should define test applied to verify the normality of distribution.

In case of parametric distribution mean ± SD should be presented, while for nonparametric distribution – median accompanied by minimum and maximum value.

Results:

In case of parametric distribution mean ± SD should be presented, while for nonparametric distribution – median accompanied by minimum and maximum value.

Authors should present tables instead of figures, to be easier to follow by readers

Cut-offs for per capita income should be justified based on a proper reference – Authors stated that it is added in line 187, but it is not

I do not understand the following sentence: „UsunÄ…Å‚bym EUR podobnie jak nie ma PLN – oba sa podane w nagÅ‚ówku. Zatem tu zbÄ™dne.”

Discussion:

Authors should present issue of fruit purchase in the aspect of buying them for specific consumers, e.g. mothers buying for children, or older members of the family (is innovativeness the only determinant?)

Authors should present issue of choice of specific fruit products – what products are chosen?

Authors should base their discussion of a proper national and international perspective

Conclusions:

Authors should briefly (2-3 sentences are enough) present major conclusions directly from their study

Authors’ Contributions:

Authors should properly define the contributions – e.g. what do Authors mean by “validation” if they did not conduct any validation in their study?

Author Response

We have provided our responses to Reviewer 2 comments in the attached document. 
